# Advances in Image-Guided Radiotherapy in the Treatment of Oral Cavity Cancer

**DOI:** 10.3390/cancers14194630

**Published:** 2022-09-23

**Authors:** Hsin-Hua Nien, Li-Ying Wang, Li-Jen Liao, Ping-Yi Lin, Chia-Yun Wu, Pei-Wei Shueng, Chen-Shuan Chung, Wu-Chia Lo, Shih-Chiang Lin, Chen-Hsi Hsieh

**Affiliations:** 1Department of Radiation Oncology, Cathay General Hospital, Taipei 106, Taiwan; 2School of Medicine, College of Medicine, Fu Jen Catholic University, New Taipei City 242, Taiwan; 3School and Graduate Institute of Physical Therapy, College of Medicine, National Taiwan University, Taipei 100, Taiwan; 4Physical Therapy Center, National Taiwan University Hospital, Taipei 100, Taiwan; 5Department of Otolaryngology Head and Neck Surgery, Far Eastern Memorial Hospital, New Taipei City 220, Taiwan; 6Head and Neck Cancer Surveillance and Research Group, Far Eastern Memorial Hospital, New Taipei City 220, Taiwan; 7Department of Electrical Engineering, Yuan Ze University, Taoyuan 320, Taiwan; 8Department of Dentistry, Far Eastern Memorial Hospital, New Taipei City 220, Taiwan; 9Division of Oncology and Hematology, Department of Internal Medicine, Far Eastern Memorial Hospital, New Taipei City 220, Taiwan; 10Division of Radiation Oncology, Department of Radiology, Far Eastern Memorial Hospital, New Taipei City 220, Taiwan; 11Faculty of Medicine, School of Medicine, National Yang Ming Chiao Tung University, Taipei 112, Taiwan; 12Division of Gastroenterology and Hepatology, Department of Internal Medicine, Far Eastern Memorial Hospital, New Taipei City 220, Taiwan; 13Ultrasonography and Endoscopy Center, Far Eastern Memorial Hospital, New Taipei City 220, Taiwan; 14Graduate Institute of Medicine, Yuan Ze University, Taoyuan 320, Taiwan; 15General Education Center, Lunghwa University of Science and Technology, Taoyuan 333, Taiwan; 16Institute of Traditional Medicine, School of Medicine, National Yang Ming Chiao Tung University, Taipei 112, Taiwan

**Keywords:** IGRT, radiotherapy, oral cavity cancer, overall survival, overall treatment time of radiotherapy

## Abstract

**Simple Summary:**

Image-guided radiotherapy (IGRT) overcomes geographic changeduring treatment and avoids underdose of the target, which provides better treatment outcomes for patients with oral cavity cancer (OCC) than intensity-modulated radiotherapy (IMRT). Additionally, IGRT also produces less radiation dose exposure to normal tissues that can cause fewer acute and late complications. All these potential benefits of IGRT can provide OCC patients with a better quality of life.

**Abstract:**

Image-guided radiotherapy (IGRT) is an advanced auxiliary radiotherapy technique. During cancer treatment, patients with oral cavity cancer (OCC) experience not only disease but also adverse effects due to RT. IGRT provides the relevant advantages of RT by precisely delivering tumoricidal doses via real-time knowledge of the target volume location and achieves maximal tumor control with minimal complications as recommended for cancer treatment. Additionally, studies have shown that IGRT can improve clinical outcomes in terms of not only treatment side effects but also survival benefits for cancer patients. IGRT can be performed alongside various imaging methods, including computed tomography and magnetic resonance imaging, and at different times during the radiotherapy regimen. This article reviews the literature to discuss the effects and importance of IGRT for patients with OCC, examines the rationale underlying the advantages of IGRT, discusses the limitations of IGRT with respect to different techniques, and summarizes the strategies and future prospects of IGRT in the treatment of OCC.

## 1. Introduction

Squamous cell carcinoma of the head and neck affects individuals throughout the world. In the United States, the disease accounts for 52,010 new cases per year, representing 4% of the incidence of all cancers [1]. In the European Union (EU), an estimated 128,600 new individuals will develop head and neck cancer [2]. In India, over 1,000,000 patients with squamous cell carcinoma of the head and neck are registered every year [3]. In Taiwan, 8204 patients were diagnosed with squamous cell carcinoma of the head and neck in 2019, representing the fifth leading cause of cancer death [4]. The primary site of the tumor is an important prognostic factor for head and neck cancer. Oral cavity cancer is associated with poorer disease control than that of other primary sites [5,6]. Lin et al., showed a high locoregional recurrence rate of up to 57% in buccal mucosa cancer patients [7]. The M.D. Anderson Cancer Center reported a 5-year survival rate of buccal mucosa cancer of only 63% [8]. The 3-year local regional recurrence-free survival rates of oropharyngeal, hypopharyngeal, and oral cavity primary cancer are 94%, 76%, and 58%, respectively [9,10].

Combined treatment is the current standard strategy for advanced oral cavity cancer. The EORTC 22931 trial surveyed treatment strategies and revealed better overall survival (OS) with surgery followed by concurrent chemoradiotherapy (CCRT) than with radiotherapy alone [11]. The RTOG 95-01 trial also showed a similar OS benefit with combined therapy [12]. Adjuvant therapy, including radiotherapy (RT) alone and with concurrent CCRT, has been shown to effectively improve local control and overall survival in head and neck cancer, especially for locally advanced disease [12,13,14,15]. Some studies have revealed that neoadjuvant CCRT is highly effective for treating tumors and is able to decrease the recurrence rate and improve overall survival in advanced oral cavity cancer [16,17].

However, RT of the oral cavity also causes adverse effects that deteriorate the patient’s quality of life, including odynophagia, mucositis, xerostomia, taste change, trismus, body weight loss, nasogastric tube dependence, and long-term side effects such as osteonecrosis of the mandible bone [18,19]. Balancing treatment-related adverse effects and survival benefits is an important issue in clinical practice. In order to improve the treatment quality, studies have been developing more advanced RT treatment techniques over the past decades. Improvements in computer calculation capacities and mechanical machine structures have led to the development of a variety of treatment techniques, ranging from two-dimensional (2D) bilateral/box irradiation and three-dimensional (3D) conformal treatment to intensity-modulated radiotherapy (IMRT) and volumetric-modulated arc therapy (VMAT). Intensity modulation and multiple-beam angle adjustment increase the conformality of target dose coverage and decrease adverse effects by limiting unnecessary critical organ irradiation [20,21,22,23,24]. Delicate treatment field sculpture enhances dose coverage, conformality, homogeneity, and normal organ protection. However, the location of the delivered dose can easily shift away from the desired target location if the patient moves significantly whilst on the treatment bed. The more delicate the treatment field design, the greater the influence the patient’s daily position changes will have on the tumor dose distribution. Six to seven weeks are needed for oral cavity cancer radiotherapy according to the clinical condition of the patient, during which an accurate daily treatment setup is needed. Treatment position accuracy is approached through various perspectives, such as immobilization techniques and in-room laser alignment. Image-guidance systems provide a direct intuitive visual tool for isocenter localization confirmation.

Can the benefits of image-guided radiotherapy (IGRT) in daily position correction transfer to real benefits such as greater OS or locoregional survival for patients with oral cavity cancer? This question remains unanswered. Here, we summarized the evidence confirming the benefits of IGRT for patients with oral cavity cancer in clinical practice.

## 2. Methods

To conduct the systemic literature review, published trials were reviewed and collected from electronic databases. The PICOTS structure was used for clinical question evaluation and search guidance. This review was registered as a project with the Center for Open Science platform (registration DOI: https://doi.org/10.17605/OSF.IO/H4DT8) (Accessed on 29 July 2022). No ethics committee approval was required.

### 2.1. Populations and Inclusion/Exclusion Criteria

Studies reporting oral cavity cancer patients who received radiotherapy were included. The description of radiotherapy included radiotherapy and proton therapy. Studies that reported head and neck cancer patients who received proton therapy and the population included oral cavity cancer patients were included. Brachytherapy was excluded from the research.

### 2.2. Intervension and Comparison

Studies that evaluated the clinical benefit of various types of image-guided radiotherapy were included. Interventions that included any type of image guidance radiotherapy system were included. Studies that reported radiotherapy with image guidance systems including two-dimensional X-ray plain film, three-dimensional computed tomography imaging, magnetic resonance imaging, and optical image guidance were included. Populations who received radiotherapy without image-guided assistance were evaluated for comparison.

### 2.3. Outcome

The evaluated clinical outcomes included disease control, treatment compliance, adverse events, and quality of life during and after radiotherapy. The end points of disease control included local control, locoregional control, disease-free survival and overall survival.

### 2.4. Timing and Setting

There were no specific criteria of timing for disease process and no specific criteria of population setting.

### 2.5. Searching Strategy

The literature search was conducted through electronic databases including PubMed, EMBASE, Cochrane, and Web of Science. Prospective and retrospective studies published between 1 January 2020 and 20 May 2022 were included in the literature review. Case reports or studies focused on animal studies were excluded. Studies that did not report clinical outcomes were excluded from the research. The results of the research finding were listed as a PRIMSA flow diagram.

### 2.6. Results

A total of 578 studies were obtained after the search. A total of 267 articles were left after duplicate studies were removed. Articles were filtered by an automation tool. A total of 173 studies were eligible with full articles. After reviewing the full articles, 57 studies were excluded due to improper participants and improper interventions. Finally, in total, 116 studies were included in our review (Figure 1).

## 3. Image-Guided Radiotherapy Techniques

IGRT for oral cavity cancer radiotherapy can be performed with various imaging modalities and presented as different image modules. The most common image-guidance modalities are two-dimensional X-ray plain-film images and three-dimensional computed tomography images. Recently, magnetic resonance imaging (MRI) has become a topic of interest in the current implementations of IGRT (Table 1).

### 3.1. Two-Dimensional (2D) X-ray Plain-Film Image Guidance

For decades, 2D plain films with megavoltage (MV) photon images have been a popular modality for visualization of bone structure since MV photons are also a source of radiotherapy. Compared with MV photon images, kilovoltage (kV) X-ray 2D plain-film images can present a much clearer outline of skull bone landmarks and assist in head and neck posture alignment. Alongside the evolution of imaging systems, onboard KV photon image-guidance systems have become standard equipment for linear accelerators.

The 2D X-ray plain-film images are typically taken as orthogonal pairs (anterior and lateral) for confirming the alignment of the bone structure. For specific treatment techniques, such as stereotactic radiosurgery (SRS) and stereotactic body radiotherapy (SBRT), a small volume target is irradiated with a high ablative radiation dose. Accuracy of the treatment position is extremely important during the treatment process. Stereoscopic imaging through room-fixed X-ray generators and detectors with intersections at the isocenter (ExacTrac^®^, Brainlab AG, Munich, Germany) can provide high-precision image guidance for SRS, SBRT, and robotic radiotherapy.

### 3.2. Three-Dimensional Computed (3D) Tomography Image Guidance

Three-dimensional volumetric images can provide more information on soft tissue organ structures for target localization than 2D X-ray plain film. The CT-on-rail system approaches the treatment position on a linear accelerator through diagnostic CT image guidance. While the treatment position is established, the couch is slid to the diagnostic CT machine first for diagnostic CT image acquisition. After position verification, the couch then slides back to the linear accelerator for radiation delivery [25]. Regarding image and treatment system integration, an MV CT imaging guidance system and a rotational IMRT system are put in one machine. For precise positioning and maximization of the treatment and protection effects, MV CT-based image guidance is mandatory for daily tomotherapy irradiation. With proper computing software for image voxel calculation and analysis, the onboard KV imaging guidance system can also provide clear CT images for daily treatment setup.

For head and neck radiotherapy, various spatial changes are noted, including the shape of the primary tumor, the involved lymph nodes, and the size of the parotid glands, which may lead to a shift in radiation dose distribution [26,27,28,29,30]. CT images can provide information on the current patient’s condition, including the relative organ position, body geometric deformation, and tumor tracking. According to the CT guidance images, individualized adaptive radiotherapy can be considered to protect the normal organs and ensure target dose coverage [31,32,33,34,35].

### 3.3. Magnetic Resonance Imaging Guidance

It can be difficult to differentiate tumors from normal tissue if the former are embedded in an organ or confluent with peripheral tissue. MRI, unlike CT, functions by using, in part, the distribution of fat and water and provides another perspective clue for tumor evaluation.

In current radiotherapy, the most popular utilization of MR images is image adaptation. Image adaptation with MR images and CT simulation images provides important information for target delineation. However, the information mainly relies on pretreatment imaging examination; real-time information is lacking during daily radiation treatment. Therefore, MR guidance systems could be a useful tool for improving assessments of the daily condition of the tumor in real time during radiotherapy. Princess Margret Hospital uses a specially designed trolley to transport patients between the MRI system and linear accelerator. The system implements an MRI scan and radiotherapy separately, completely avoiding electromagnetic cross-interference [36]. Other systems integrating MR scanners and radiation delivery systems encounter complex electromagnetic cross-interference but solve the issue through unique designs for the magnetic field direction, shielding, and dose calculation compensation. The basis and technique for one example, the MRI-Linac system, have gradually been improved in recent years and commercialized for patient treatment [37,38,39,40,41,42].

### 3.4. Other Imaging Guidance Techniques

Infrared markers, body surface, and body temperature are also available for image guidance. Optical tracking systems focus on tracking the patient’s position through active fiducial markers, passive infrared markers, and the condition of the body surface. For frameless stereotaxis, an array with infrared light-emitting diodes is placed upon the patient’s face and fixed through a bite plate. An infrared camera captures the image of the positions of the diodes to provide positioning guidance [43,44,45,46,47]. Optical surface guidance, another guidance technique, captures the surface shape of the target area for position verification [45,48,49]. Thermal signatures are a relatively new approach for wide range position tracking [50,51].

## 4. Benefits of IGRT for OCC Patients

### 4.1. Benefits in Overall Survival and Locoregional Control

IMRT has advantages including the use of intensity-modulated beams and backward calculation. Compared with 3D conformal radiotherapy, IMRT has been shown to be a more feasible treatment technique for oral cavity cancers [52]. The efficacy in toxicity control and equivalent local control have been investigated and demonstrated to have noninferior and even better results [53,54,55,56,57,58]. Regarding protection of organs at risk (OARs), studies have revealed that IMRT can achieve better parotid gland protection without compromising the local control rate [24,58]. Furthermore, Table 2 list the investigations comparing IMRT and conventional 3D radiotherapy that have revealed significantly better locoregional control (LRC), disease-free survival (DFS), and OS with IMRT in advanced head and neck cancer treatment [59,60,61].

Compared with IMRT, helical tomotherapy (HT) has been revealed to provide better clinical outcomes, including OS, DFS, locoregional progression-free survival (LRPFS), and distant metastasis-free survival in head and neck cancer radiotherapy [62]. Hsieh et al., demonstrated significantly higher 5-year OS (86.7% vs. 47.5%) and local progression-free survival (LPFS) (85.2% vs. 58.4%) with HT than with IMRT [63] (Figure 2).

**Table 2 cancers-14-04630-t002:** Treatment outcomes with different radiotherapy techniques and forms of image guidance.

Selected Published Series	Number of Postoperative Patients	Modality	Follow-Up Period	OS	DFS	LR PF	DMF
Chen AM et al. [58]	78 (OCC: 30)	2DRT	3 years	69%	-	70%	66%
	52 (OCC: 25)	IMRT	3 years	72%	-	73%	70%
Wang ZH et al. [64]	44 (OCC: 38)	2DRT	4 years	56.8%	52.3%	-	-
	44 (OCC: 39)	IMRT	4 years	70.5%	68.2%	-	-
Chen PY et al. [60]	42	2DRT	3 years	51.2%	47.8%	53.5%	-
	72	IMRT	3 years	69.4%	70.0%	76.3%	-
Yao et al. [65]	55	IMRT	2 years	68%	74%	82%	89%
Gomez et al. [52]	35	IMRT	3 years	74%	64%	77%	85%
Chen WC et al. [57]	27	2DRT	3 years	77%	66%	-	-
	22	IMRT	3 years	67%	64%	-	-
Lin CS et al. [61]	91	2DRT	5 years	30.0%	-	30.0%	-
	83	IMRT	5 years	53.5%	-	40.5%	-
Hoffmann M et al. [66]	18	IMRT	5 years	77%	72%	78%	80%
EORTC 22931 [11]	167 (OCC: 41)	CCRT	5 years	53%		47%	
RTOG 9501 [12]	206 (OCC: 50)	CCRT	2 years			82%	
RTOG 9501 [15]	50 (206)	CCRT	5 years	46%	30%		
Hsieh et al. [63]	79	IMRT	5 years	48%	39%	58%	83%
	73	IG-IMRT (HT)		87%	74%	85%	80%

2DRT: conventional radiotherapy; OCC: oral cavity cancer; IMRT: intensity-modulated radiotherapy; IG-IMRT: image-guided intensity-modulated radiotherapy; OS: overall survival; DFS: disease-free survival; LRPF: locoregional progression-free; DMF: distant metastasis-free; CCRT: concurrent chemoradiotherapy.

### 4.2. Benefits for High-Risk Patients

The presence of lymph node metastases and extracapsular lymph nodes are correlated with a poor 5-year survival rate [8]. The European Organization for Research and Treatment of Cancer (EORTC) recognizes T3, T4, level 4/level 5 lymph node metastases, present perineural invasion (PNI), and vascular embolism as risk factors for oral cavity cancer [11]. The Radiation Therapy Oncology Group (RTOG) identified the risk factors for oral cavity cancer as having more than two metastatic lymph nodes, extracapsular extension (ECE), and a positive surgical margin [12]. For patients with locally advanced head and neck cancer, a poorer prognosis is suspected, and aggressive treatment is suggested [11,12,14,15]. Daily image-guided IMRT has been demonstrated to provide a significantly better 5-year OS and LPFS in patient groups with positive resection margins, ECE, perineural invasion (PNI), the presence of lymphovascular space involvement, more than two metastatic lymph nodes, and T3/T4 [63].

### 4.3. Benefits in Treatment Compliance

The total treatment time from surgery to completion of radiotherapy, also called the package of treatment time (POTT), also affects clinical outcomes directly. In the study by Ang et al., a POTT > 13 weeks was associated with a significantly lower LRC than a POTT ≤ 13 weeks [67]. A POTT of less than 100 days was suggested by several studies as predicting a significantly better LRC in high-risk patients and recognized as an independent predictor of OS [68,69,70]. In further time-section analysis, a prolonged time lag between operation and adjuvant radiotherapy was demonstrated to be detrimental for LRC [67,71]. Other investigations have recommend a shorter overall treatment time of radiotherapy (OTTRT) for better LRC [67,72,73,74]. Treatment interruptions of more than 10 days during radiotherapy were associated with 10–20% of 5-year recurrence-free survival [73]. Fujiwara et al., suggested an OTTRT of less than 54 days for significantly better OS [75].

IGRT provided significantly higher OS and LPFS than non-IG IMRT under similar POTTs (≤13 weeks) and OTTRTs (≤8 weeks) [63]. Further regarding the OTTRT, Muriel et al. showed that only 39% of patients completed radiotherapy within 55 days [71]. Only 52% of patients achieved an OTTRT of less than 8 weeks in the study by Langendijk et al. [74]. In contrast, image-guided IMRT results in a relatively high percentage of patients achieving an OTTRT ≤ 8 weeks. One study revealed that a higher percentage of patients had longer OTTRTs of ≥7 weeks and ≥8 weeks in the IMRT group than in the image-guided IMRT group (58.2% vs. 40% and 32% vs. 11%, respectively) [63,76] (Table 2).

## 5. Contributing Factors

### 5.1. Increased Marginal Failure Control

For patients who receive IMRT, the locoregional recurrence rate is approximately 24.1~45.4% across different studies [8,77,78,79]. Previous studies have shown that marginal failure can account for up to 83% of total locoregional failure in IMRT with only a 3 mm PTV margin and 15.6–53% in IMRT with a 5 mm PTV margin [58,63,65,78,80,81,82,83]. Saha et al., reviewed head and neck cancer patients, calculated the setup errors, and suggested required PTV margins of 5.7 mm, 5.3 mm, and 6.2 mm in the *x*-, *y*-, and *z*-axes, respectively, for non-IG IMRT [84]. Zeidan et al., reported that IGRT can improve residual setup errors; however, 29% residual errors >3 mm were still noted during head and neck radiotherapy treatment, even with IGRT every other day [85]. A sufficient PTV margin is necessary for IMRT treatment without IGRT (Table 3).

With the assistance of IGRT, Chen et al. [86,87] revealed no specific difference in locoregional control between the 5 mm and 3 mm PTV margin groups, with marginal recurrence rates of 7% and 5%, respectively. Farrag et al. [88] reported a marginal failure rate of 3.2% with IGRT assistance. Hsieh et al. [89] reported no marginal failure, and only a 4% outfield failure rate with a 3 mm PTV margin in IG-IMRT. Hsieh et al., compared 79 patients treated with 5 mm PTV non-IG IMRT and 3 mm PTV IGRT. The local regional failure rate was 24% for non-IG IMRT and 6.8% for IGRT. The marginal and outfield failure rates were 16.5% for non-IG IMRT (13/79) and less than 5% (1/73) for IG-IMRT. The proportion of marginal to total locoregional failure was 52.5% for non-IG IMRT, while no marginal failure was noted in the IG-IMRT group. IGRT had a significantly better 5-year LPFS and OS than IMRT alone [63]. IGRT can modify the failure pattern for head and neck cancer radiotherapy and is a valuable tool for improving locoregional control in IMRT [80,89].

### 5.2. Reduced Treatment Side Effects

Most adverse effects that occur during radiotherapy are mucositis, dermatitis, xerostomia, and trismus (Table 4). Discomfort during treatment can lead to malnutrition and body weight loss, and a feeding tube may be required for nutritional maintenance [5,77,90]. Capuano et al., revealed that a body weight loss of more than 20% of prediagnosis weight during CCRT is significantly correlated with a poor survival rate [91]. IGRT allows PTV margin shrinkage by daily adjustment that could reduce the total treatment volume and possibly minimize the adverse effects. Hsieh et al. revealed that IGRT helped to significantly reduce grade 2/3 body weight loss compared with non-IG IMRT [63]. To improve treatment quality and reduce survival influence, efforts to avoid the adverse effects of treatment should be prioritized.

### 5.3. Xerostomia

Xerostomia is a common adverse effect of head and neck cancer radiotherapy and is important since it can decrease oral hygiene and increase the possibility of carious teeth and periodontal disease [21,77,92,93]. Relative stimulated saliva values at 6 months after the completion of radiotherapy are associated with quality of life [94]. Multiple performance metrics, including eating, speaking, communication, pain, sleep, and emotion, were revealed to be correlated with stimulated and unstimulated saliva flow [95,96,97,98]. Previous studies have shown that grade 3 xerostomia was reported in approximately 11.5–58% of patients who underwent IMRT [5,23,90,99]. Nguyen et al., showed that IG-IMRT can achieve significantly better parotid gland protection, including the mean dose and V_40_, through a highly demanding dose constraint [100]. With IG-IMRT, xerostomia can generally be maintained mainly at grade 2, while grade 3 xerostomia is rare [63,76,87,89]. Additionally, the rates of patients treated with IGRT experiencing grade 2 late xerostomia at 12 months and 24 months were 18% and 10%, respectively, which showed a trend of decreasing late toxicities to the salivary gland [89].

### 5.4. Posttreatment Esophageal Stricture and Gastrostomy Tube Dependence

With daily IGRT, the PTV margin can be reduced to 3 mm from the OAR without compromising locoregional control [63,80,89]. A percutaneous endoscopic gastrostomy tube was needed in approximately 15.9–24% of head and neck cancer patients who received concurrent chemoradiotherapy with IMRT [78,90]. With IGRT assistance, the 1-year actuarial gastrostomy tube-free survival can be increased from 81% to 90%, and the required posttreatment esophageal stricture dilatation can be significantly reduced from 14% to 7% with a reduction in the PTV margin [86].

Compared with IMRT alone, IGRT also reduced the incidence of leukopenia and thrombocytopenia [63]. A case of trismus recovered steadily over time after treatment with image-guided IMRT [89]. Hsieh et al., reviewed the clinical treatment outcomes of oral cavity cancer patients with image-guided IMRT and reported acceptable adverse effects and good treatment compliance, concurrent with those of chemotherapy [63,76,87,89].

### 5.5. Performance Status and Quality of Life

For head and neck radiotherapy quality of life (QOL) evaluation, popular questionnaires are the EORTC Quality of Life Questionnaire-C30 (QLQ-C30) score, the disease-specific EORTC QLQ module for head and neck cancer (QLQ-H&N35), the head and neck quality of life questionnaire (NHQOL), and the EuroQol five-dimensional questionnaire (EQ-5D) [101,102,103]. Jabbari et al., investigated the quality of life of head and neck cancer patients who received radiotherapy. The xerostomia questionnaire (XQ) showed a gradual improvement in the xerostomia over time, with a median XQ score of 32 for patients who received IMRT at 12 months after irradiation [21]. The NHQOL also revealed gradual improvements at 6 months post-irradiation. Lin et al. reported a median NHQOL score of 17 for IMRT at 12 months postradiotherapy, a recovery to approximately 50% of the original QOL total score before radiotherapy [96]. Voordeckers et al. utilized the EORTC QLQ-C30 to evaluate head and neck radiotherapy with IGRT. Their results revealed an improved functional, global health, and symptom score of approximately 90% at 12 months after radiotherapy, which subsequently reached the baseline value 18 months after irradiation [104]. The daily treatment position accuracy improvement by IGRT can assist in the maintenance and recovery of the quality of life of patients after radiotherapy.

## 6. Limitations of IGRT

IGRT has benefits as mentioned above, however, there are several limitations for IGRT for clinical applications and we tried to discuss these limitations in the following paragraph (Table 5).

### 6.1. Extra Time for Imaging Guidance

For delicate image guidance such as 3D CT-based image guidance and MR image guidance, extra time is required for image generation. Patients need to stay in a stable position during the whole process, including image guidance, post-image guidance position adjustment, and the whole radiation delivery process. Patients experiencing severe bone pain may find maintaining their treatment posture for extended periods of time difficult. More safety concerns need to be addressed for patients experiencing frequent cough with sputum and dyspnea with oxygen support, especially while in the treatment room. In such situations, image guidance may not be the best choice for the patient.

### 6.2. Intrafraction Motion

IGRT can be performed before radiotherapy or during radiotherapy. For IGRT before radiotherapy, intrafraction motion should be kept in mind [105,106]. Champion et al., showed that tumors of the head and neck can move independently from the bone structures due to breathing, swallowing, and coughing [107]. The treatment should be completed as soon as possible after position adjustment after IGRT. For IGRT during radiotherapy, such as dynamic MRI, different imaging principles can lead to subtle differences in size and tumor region. The registration of the tumor area between the original CT contouring and MR signal in IGRT and the decision to interrupt the treatment due to bulk motion highly relies on the physician’s experience.

### 6.3. Comparison between CT-Guided and MR-Guided Radiotherapy

Compared with CT-guided radiotherapy, MR-guided radiotherapy does not involve ionizing radiation and is advantageous in providing different imaging modalities, such as diffusion-weighted imaging and perfusion-weighted imaging, for differentiating tumors from the surrounding tissue [108,109,110,111,112,113]. Current MR-guided imaging systems also provide adaptative planning functions [114,115,116,117]. Dynamic MR-guided, real-time intrafraction tumor motion tracking is available during radiotherapy, and structural deformation during treatment can be clearly visualized under MR image guidance. However, MR-guided radiotherapy faces several challenges. The extra magnetic field affects the electron traveling tract and interferes with the dose distribution. Sophisticated dose calculations are required for addressing the electron return effect and re-entrance effects caused by the Lorentz force [118,119,120]. The dose deposition shift in a magnetic field is complicated and determined by the magnetic field force, field size, tissue density, and geometry. Several studies have investigated the dose distribution effect and have built algorithms for specific dose calculations [118,121,122,123,124]. Additionally, the MR image guidance system must balance the MR image quality with the time consumption for image generation and electron trajectory affected by the magnetic field strength [125]. In addition, a possible restricted treatment field length has been reported [126] and the noise of the MR machine may be disturbing to the patient [127]. Online adaptive plans require dozens of minutes and are not suitable for patients who cannot remain still for extended periods of time.

### 6.4. Application of IGRT in Proton Therapy

Proton therapy is characterized by a specific, physical dose distribution of proton particles and is suitable for both primary treatment and reirradiation for recurrent cancer [128,129,130]. The dose increases at the end of the particle range and decreases sharply after the Bragg peak, contributing to protecting the OARs [131,132]. Clinical evaluations have shown that proton therapy causes less treatment toxicity with no significant difference in survival than conventional radiotherapy [133,134,135]. Intensity-modulated proton therapy is recommended for head and neck cancer treatment [136] and is more sensitive to body deformation and organ motion than conventional photon radiotherapy [137]. In cases of dose deviation, adaptive planning has been suggested [138,139,140]. Treatment position accuracy plays a very important role in proton therapy. Currently, most in-room image guidance systems for proton therapy rely mainly on 2D orthogonal X-ray images followed by tomography imaging guidance systems [141,142,143], while IGRT can assist in the treatment position accuracy [144]. Several studies have investigated proton dose recalculation and adaptive proton plans based on cone-beam CT images [145,146,147]. MR image guidance systems may interfere with the dose distribution in proton therapy [148,149]. After proper correction, the MR image guidance system provides tumor location acquisition, further anatomic structure deformation evaluation, tumor biology and heterogeneity assessment, and treatment quality improvement for proton therapy [143,150].

## 7. Future Prospects

### 7.1. Cooperation between IGRT and PET-CT

PET-CT with different isotope recombination schemes can provide metabolic images for tumor activity and reference hypoxia regions [151,152,153,154]. The hypoxic region in tumors has a higher radioresistance potential and is associated with a poor prognosis in head and neck cancer [155,156,157]. The metabolic tumor volume and the total lesion glycolysis of the tumor are prognostic factors of OS [158]. Registration of PET-CT to radiotherapy contouring is a feasible strategy. However, anatomical changes in organs and tumors have been documented during radiotherapy [26,27,29]. The pretreatment IMRT plan and PET-CT may only indicate possible hypoxia and high activity regions of the tumor at the beginning of treatment. Since the outline shape and biological characteristics of the tumor may change during treatment, several studies have demonstrated that adaptive dose painting during radiotherapy according to PET-CT is acceptable [159,160]. IGRT can be used to instantly evaluate tumor status during radiotherapy. Information sharing between IGRT and PET-CT may improve treatment quality and disease control. Long-term clinical outcome evaluation is needed to compare the benefits from IMRT with PET-CT and IGRT with PET-CT.

### 7.2. IGRT with Dose Painting for Biologic Mapping

Both 3D CT-based and MR image guidance systems are available for radiotherapy. An image-guidance system can help physicians achieve tumor response during radiotherapy. The pretreatment apparent diffusion coefficient (ADC) value of diffusion-weighted imaging is correlated with the recurrence rate [161,162,163,164] and the intratreatment tumor volume is associated with local tumor control [165]. Other diagnostic imaging tools, such as PET-CT, can provide biologic imaging information, including tumor activity and identification of hypoxic regions [151,153]. MRI receives and analyzes signals from the spin–spin relaxation time and spin-lattice relaxation of atomic nuclei and surrounding structures. Hydrogen is not the only element that can be detected through MR imaging; multinuclear imaging can be performed with proper parameter adjustment to provide more information about the specific desired target distribution. Adaptive radiotherapy includes anatomic-adapted adaptive radiotherapy and response-adapted adaptive radiotherapy [166]. Incorporative imaging information provides the opportunity to understand tumor changes, including volume, perfusion, and other imaging biological markers during treatment. Dose painting with an escalation dose on a resistant tumor region is feasible [159,160,167]. A biologic adaptive plan can possibly further improve disease control, and good efficacy is anticipated with further investigations.

### 7.3. IGRT for Reirradiation

Reirradiation is always a challenge in cancer management. Patients who experience recurrence or new lesions after radiotherapy may benefit from reirradiation [168]. Reirradiation can be performed with 3D CRT [169,170,171], IMRT [168,169,170,172,173,174], and SBRT [170,175,176,177,178]. Increasing the reirradiation dose is an independent factor for OS, and the posttreatment tumor response is strongly correlated with LRPFS and OS [169,172,179,180]. However, due to normal tissue tolerances, the reirradiation dose that can be delivered is limited, and treatment-related toxicity is an important concern [179]. More than one-third of patients may experience grade 3–4 toxicity and even treatment-related mortality [170,173,174,181]. Increasing the PTV margin may reduce the failure rate but also increase toxicity [182,183]. IGRT is an effective way to reduce the PTV margin for reducing treatment toxicity while protecting the OARs with further escalation of the effective treatment dose and improving the OS [46,184,185].

### 7.4. SBRT in Oral Cavity Cancer

SBRT has been demonstrated as a feasible treatment strategy for recurrent cancer [170,175,176,177,178,186,187]. In addition to recurrent tumors, SBRT has also been applied in the treatment of primary tumors [188]. Early-stage oral cancer tumors are usually small and require surgery only. Re-surgery is suggested for the involved surgical margin. However, for patients unsuitable for further operation, SBRT can be an alternative treatment choice [189]. For small, precise treatment fields, IGRT plays an important role in accurate SBRT for enhancing tumor control through high-dose delivery.

## 8. Conclusions

IGRT results in better overall survival and local progression-free survival for patients with locally advanced OCC that might be contributed to by the better target volume dose conformity and homogeneity and allows for the daily adjustment in setup error to overcome the marginal failures. Additionally, daily adjustment allows PTV margin shrinkage that could reduce the total treatment volume and minimize the exposure of normal tissues to radiation. Therefore, patients treated with IGRT might have a lower toxicity and a better chance of completing OTTRT within the suggested treatment durationand hence a better overall outcome. Moreover, less radiation dose exposure to normal tissues can cause fewer acute and late complications that also provide OCC patients with a better quality of life (Figure 3). The current review provides insight into the potential benefits of IGRT which can be incorporated into our current understanding for future treatment.

## Figures and Tables

**Figure 1 cancers-14-04630-f001:**
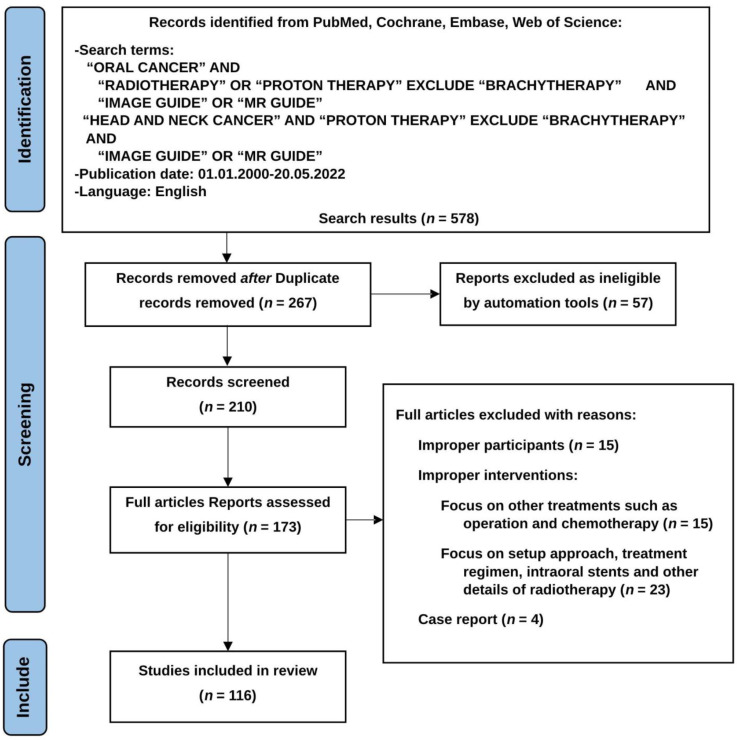
To conduct the systemic literature review, published trials were reviewed and collected from electronic databases. The PICOTS structure was used for clinical question evaluation and search guidance.

**Figure 2 cancers-14-04630-f002:**
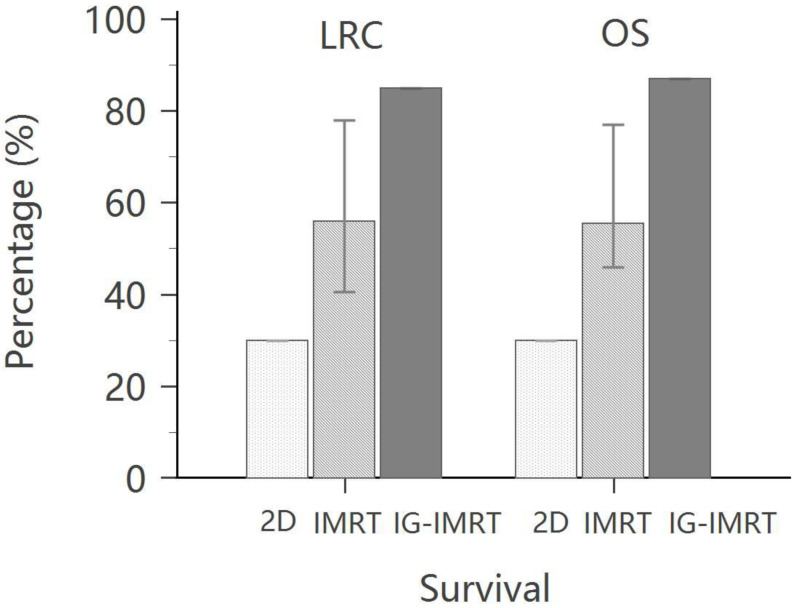
The comparison of local regional control rate and overall survival rate between 2D, intensity-modulated radiotherapy (IMRT) and image-guided (IG)-IMRT for patients with head and neck cancer.

**Figure 3 cancers-14-04630-f003:**
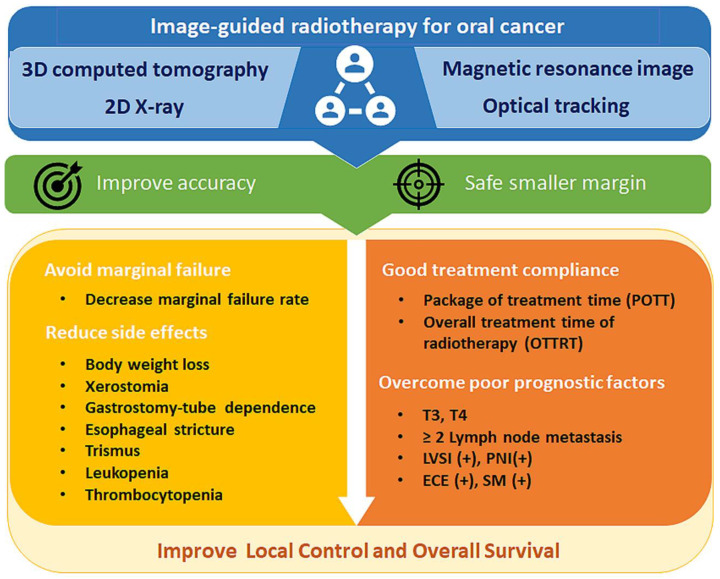
IGRT decreases toxicities, increases compliance, and overcomes the marginal failures that contribute to a better overall outcome and provide OCC patients with better quality of life.

**Table 1 cancers-14-04630-t001:** Comparison advantages and disadvantages among different image modalities.

	Landmarks	Dimensionality	Advantages	Disadvantages
Two-dimensional X-ray plain-film imaging guidance	Bone structure	2D	Fast Less intrafractional motion	Unable to observe soft tissue change Cannot verify position with three-dimensional structures
Three-dimensional computed tomography imaging guidance	Soft tissue and bone structure	3D	Verify position with three-dimensional structures Can closely observe spatial change during radiotherapy Adaptive plan according to patient’s condition	Unable to differentiate tumors from nearly soft tissue
Magnetic resonance imaging guidance	Water and fat distribution	3D	Verify position with three-dimensional structures Can closely observe spatial change during radiotherapy Can differentiate tumor from nearly soft tissue clearly Adaptive plan according to patient’s condition	Magnetic field will interfere with electron trajectory and dose distribution Acquired experience to register MR image to previous CT images Machine noise
Infrared markers for image guidance	Infrared marker position	others	Verify position with infrared markers positions. Infrared markers can be placed at the points we desired and interested.	The motion of infrared markers is highly affected by breathing and can easily trigger system inaccurate position recognition and may increase the frequency of treatment interruption and treatment duration
Body surface for image guidance	Surface shape	others	Verify position with all points position on specified body surface. The geometric surface of the body surface is delicate, which can increase the setup accuracy	The geometric surface of the body surface is delicate and complicate which triggers system inaccurate position recognition easily and may increase the frequency of treatment interruption and treatment duration
Body temperature for image guidance	Body temperature mapping	others	Verify position with body temperature mapping	Body temperature may change according to different conditions, which is under investigation

2D: two dimensions; 3D: three dimensions.

**Table 3 cancers-14-04630-t003:** Marginal failure rate with intensity-modulated radiotherapy and image-guided intensity-modulated radiotherapy.

	Modality	Number of Enrolled Patients	Percentage of Oral Cavity Cancer	Margin of PTV	No. of Marginal Failures/No. of Locoregional Failures	Percentage of Marginal Failure
Bern University Hospital, Switzerland [81]	IMRT	53	100%	3 mm	10/12	83%
University of Iowa Health Care, USA [65]	IMRT	49	100%	5 mm	2/8	25%
Princess Margaret Hospital, Canada [82]	IMRT	180	100%	-	12/38	32%
University of California Davis School of Medicine, Canada [80]	IMRT	90	48%	3–5 mm	6/17	35%
University of California Davis School of Medicine, Canada [58]	IMRT	52	48%	3–5 mm	4/13	31%
Stanford University Medical Center, Canada [78]	IMRT	30	100%	3–5 mm	2/11	18%
National Cancer Center Hospital East, Japan [83]	IMRT	122	48%	5 mm	5/32	16%
Far Eastern Memorial Hospital, Taiwan [63]	IMRT	79	100%	5 mm	10/19	53%
Far Eastern Memorial Hospital, Taiwan [63]	IG-IMRT	73	100%	3 mm	0/5	0%
University of California Davis Comprehensive Cancer Center, USA [86]	IG-IMRT	103	31%	5 mm	5/76	7%
University of California Davis Comprehensive Cancer Center, USA [86]	IG-IMRT	264	21%	3 mm	4/76	5%
University of California Davis Comprehensive Cancer Center, USA [86]	IG-IMRT	367 (103–5 mm, 264–3 mm)	24%	3–5 mm	9/76 (5–5 mm, 4–3 mm)	12%

PTV: planning target volume; No.: number; IMRT: intensity-modulated radiotherapy; IG-IMRT: image-guided intensity-modulated radiotherapy.

**Table 4 cancers-14-04630-t004:** Treatment-related adverse effects with intensity-modulated radiotherapy and image-guided intensity-modulated radiotherapy.

Side Effect	Modality	Number of Enrolled Patients	Percentage of Oral Cavity Cancer	Gr.1	Gr.2	Gr.3	Gr.4	Significance
Body weight loss								
Hsieh et al. [63]	IMRT	79	100%	51	27	1	0	
	IG-IMRT	73	100%	62	11	0	0	*p* = 0.004
Xerostomia								
Chen PY et al. [60]	CRT	42	100%	-	10 (34.5%)	0	-	
Chen WC et al. [57]	CRT	27	100%	-	−82%		-	
	IMRT	22	100%	-	−36%		-	
Chen PY et al. [60]	IMRT	72	100%	-	8 (14.0%)	0		
Moon et al. [5]	IMRT	51	45.1%	-	-	10 (19.6%)	-	
Wang et al. [23]	IMRT	26	92.3%	-	-	3 (11.5%)	-	
Seung et al. [90]	IMRT	69	26%	0	29	40 (58%)	0	
Hsieh et al. [76]	IG-IMRT	19	100%	10	9	0	0	
Hsieh et al. [89]	IG-IMRT	53	100%	(66.7%)	(33.3%)	0	0	
Leucopenia								
Hsieh et al. [63]	IMRT	79	100%	49	9	5	2	
	IG-IMRT	73	100%	25	17	6	1	*p* = 0.007
Thrombocytopenia								
Hsieh et al. [63]	IMRT	79	100%	59	3	2	0	
	IG-IMRT	73	100%	41	1	0	0	*p* = 0.003

IMRT: intensity-modulated radiotherapy; IG-IMRT: image-guided intensity-modulated radiotherapy.

**Table 5 cancers-14-04630-t005:** The pros and cons for image-guided radiotherapy for patients with oral cavity cancer.

Modality	Pros	Cons
Image-guided radiotherapy	Improve set-up accuracy Safe smaller margin -Clinical benefits including avoid marginal failure, reduce treatment side effects, good treatment tolerance, and overcome poor prognostic factors Understand tumor condition during treatment period Online adaptive plan Opportunities of biological mapping for dose painting.	Increase immobility duration -Not suitable for patients with claustrophobia, unstable conditions, or severe painIncrease ionizing radiation exposure (2D, 3D CT image) Disturbing noise during image scanning (MRI) The registration between the original CT contouring and the IGRT images and the decisions to interrupt the treatment due to bulk motion highly relies on the physician’s experience.

2D: two-dimensional; 3D: three-dimensional; CT: computed tomography; MRI: magnetic resonance imaging; IGRT: image-guided radiotherapy.

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
