# Peer review of "Advances in Image-Guided Radiotherapy in the Treatment of Oral Cavity Cancer"

_cancers, 2022, doi:10.3390/cancers14194630_

Round 1

Reviewer 1 Report

It was a great pleasure to review this well-written and comprehensive paper. The authors provide adequate context to highlight the significance of the topic, and their review of existing options as well as current status and view to the future are balanced and comprehensive. Outstanding paper!

Author Response

Professor Davina Wang,                       September 17, 2021

Editor,

Cancers

Dear Professor Wang:

On behalf of all authors, I appreciate the time and effort of the editor and reviewers in critiquing our work (cancers-1865778). Attached is the point-by-point response to the reviewers’ comments. We re-submit this manuscript for re-consideration for publication in Cancers. Thank you for your great effort on our work.

Yours sincerely,

Chen-Hsi Hsieh, M.D., Ph.D.

Division of Radiation Oncology, Department of Radiology,

Far Eastern Memorial Hospital,

No.21, Sec. 2, Nanya S. Rd., Banciao Dist., New Taipei City 220, Taiwan

Fax: 886-2- 8966-0906

Phone: 886-2-8966-7000 ext. 1033

Major compulsory revisions:

Reviewer 1

It was a great pleasure to review this well-written and comprehensive paper. The authors provide adequate context to highlight the significance of the topic, and their review of existing options as well as current status and view to the future are balanced and comprehensive. Outstanding paper!

Response: Thank you for your time and appreciate for your comments.

Reviewer 2 Report

The Authors outlined, within a comprehensive and interesting review, the current knowledge on image guided radiotherapy for oral cavity tumours. The manuscript is of interest. Few comments below.

1)     I feel the manuscript has been organized and drafted mostly as a qualitative review, which is fine. I would, however, try to improve the methodology employed, making explicit the choice undertaken during the draft of the manuscript.

2)     Please refer to the PICOTS criteria and frame populations, intervention(s), comparator(s), outcomes, timing, setting.

3)     Please provide details of the search strategy employed in the present review paper.

4)     I would suggest referring to the PRISMA criteria (please add a PRISMA flow-chart).

5)     I would suggest reducing the textual part and add tables to wrap up the main findings and help the reader navigating the data presented.

Author Response

Attached was the point by point response for referee's suggestions !

Round 2

Reviewer 2 Report

I do not have any further comment.